# FROM UNIFORM TO ADAPTIVE: GENERAL SKIP-BLOCK MECHANISMS FOR EFFICIENT PDE NEURAL OPERATORS

## ABSTRACT

In recent years, Neural Operators(NO) have gradually emerged as a popular approach for solving Partial Differential Equations (PDEs). However, their application to large-scale engineering tasks suffers from significant computational overhead. And the fact that current models impose a uniform computational cost while physical fields exhibit vastly different complexities constitutes a fundamental mismatch, which is the root of this inefficiency. For instance, in turbulence flows, intricate vortex regions require deeper network processing compared to stable flows. To address this, we introduce a framework: Skip-Block Routing (SBR), a general framework designed for Transformer-based neural operators, capable of being integrated into their multi-layer architectures. First, SBR uses a routing mechanism to learn the complexity and ranking of tokens, which is then applied during inference. Then, in later layers, it decides how many tokens are passed forward based on this ranking. This way, the model focuses more processing capacity on the tokens that are more complex. Experiments demonstrate that SBR is a general framework that seamlessly integrates into various neural operators. Our method reduces computational cost by approximately 50% in terms of Floating Point Operations (FLOPs), while still delivering up to $2\times$ faster inference without sacrificing accuracy.

## 1 INTRODUCTION

The ability to solve Partial Differential Equations (PDEs) is fundamental to modern science and engineering, underpinning advances from climate modeling to aerospace design Chen et al. (2023); Bi et al. (2022); Palmer (2019); Wu et al. (2024).

Concurrently, a powerful class of deep learning models, Transformer-based Neural Operators(NO), has seen a surge in development, rapidly becoming a dominant approach in the field of PDEs solving. Influential works such as OFormer Li et al. (2022), GNOT Hao et al. (2023), and Transolver Wu et al. (2024) exemplify this trend, demonstrating commendable accuracy in modelling complex physical systems.

However, its practical utility becomes questionable in engineering tasks that demand repeated model executions or large-scale computations. For example, in the thermal design of semiconductor chips, a single PDE solver might be called thousands of times. This repetitive execution turns even modest per-inference inefficiencies into a prohibitive computational barrier, making them incompatible with the agile design cycles of modern industry.

The reason for this is that these approaches enforce a uniform, full-depth computational process across the entire problem domain. This means that markedly different physical regions are treated as if they were identically complex. This core inefficiency is not an isolated issue but prevails across PDEs solving, from identifying stress hotspots in materials science to modeling shockwaves in aerodynamics. For instance, in a climate simulation, a vast and stable oceanic region is processed with the same deep neural architecture as a rapidly evolving hurricane—a strategy that squanders immense computational resources on areas that are inherently simple to predict.

To address this fundamental inefficiency, we argue for a paradigm shift away from the rigid, one-size-fits-all approach. We propose a new, adaptive principle for neural operators: the computational resources allocated to a region, particularly network processing depth, should be directly proportional to the physical complexity exhibited within that region. This ensures that the model's effort is intelligently focused only where it is needed most.

To implement this principle, we introduce SBR, a novel and general framework designed as a modular enhancement for existing Transformer-based neural operators. The SBR framework is composed of two core components.

First, a global router module performs a one-time, upfront analysis of the entire input domain. Its function is to generate a static complexity ranking for all discretized tokens, effectively creating a fixed "computational priority plan" that guides the entire forward pass. Second, an adaptive processing backbone utilizes the previously computed "computational priority plan" to dynamically adjust the number of active tokens at different depths of the network. In the shallower layers, a wider set of tokens is processed to capture global context. As the network deepens, this backbone progressively narrows its focus, concentrating the deep, computationally intensive transformations only on the subset of tokens identified as most critical. This entire framework is designed to be end-to-end differentiable, allowing the router's importance assessment to be learned directly from the final PDEs solution task.

The main contributions of this work are summarized as follows:

- We are the first to systematically identify and formulate the "uniform computation" bottleneck in existing neural operators, revealing the fundamental mismatch between their rigid computational strategy and the heterogeneous complexity of physical fields.

- We propose Skip-Block Routing (SBR), a novel and general framework that introduces a new adaptive paradigm to neural operators. SBR is founded on the principle of decoupling the assessment of complexity from the main computational pipeline.

- The SBR framework is lightweight and efficient, built upon a static, importance-based routing mechanism that enables a hardware-friendly gather-and-scatter execution flow. This design ensures that SBR is easily trainable with standard gradient-based methods, without requiring reinforcement learning.

- We conduct extensive experiments on a range of PDE benchmarks, demonstrating that SBR can be seamlessly integrated into various operators. Our results show that SBR fundamentally improves efficiency by reducing the computational cost by approximately 50% in terms of FLOPs. This core reduction in workload translates directly to practical performance benefits, enabling up to 2x faster end-to-end inference without sacrificing predictive accuracy.

## 2 RELATED WORK

### 2.1 NEURAL OPERATORS FOR PDE SOLVING

Neural Operators (NOs) have emerged as a powerful paradigm for data-driven PDE solving, learning the mapping between function spaces to achieve orders-of-magnitude faster inference than traditional numerical methods. Foundational approaches like the kernel-based DeepONet Lu et al. (2019) and the influential Fourier Neural Operator (FNO) Li et al. (2020) established this paradigm by approximating the solution operator via basis expansions or transformations in the frequency domain.

More recently, the field has gravitated towards Transformer-based architectures as the state-of-the-art. This line of research, initiated by models like OFormer Li et al. (2022) and later advanced by the general-purpose GNOT Hao et al. (2023), IPOT Lee & Oh (2024) and the physics-aware Transolver Wu et al. (2024), leverages the powerful self-attention mechanism. By treating the discretized domain as a sequence of tokens, these models have demonstrated remarkable accuracy in capturing complex, long-range physical dependencies.

However, the strength of these Transformer-based operators is intrinsically tied to a design that enforces uniform computation. Their fundamental working principle is that to understand the state

of any single point, the model must consider its relationship with every other point in the entire system. This global interaction is applied indiscriminately: a point in a stable, quiescent region is forced to undergo the same expensive computational process of interacting with all other points as a point within a critical vortex. This inherent, one-size-fits-all approach to modeling physical interactions is the central bottleneck that our work aims to address.

## 2.2 ADAPTIVE COMPUTATION FOR EFFICIENT DEEP LEARNING

The challenge of computational inefficiency has been extensively studied in deep learning, motivating various adaptive computation techniques. Early strategies like early exiting Teerapittayanon et al. (2016) and token pruning Rao et al. (2021), primarily explored in CV and NLP, rely on local, layer-by-layer dynamic decisions, but often at the risk of suboptimal choices or irreversible information loss.

A key advancement in this area is Mixture-of-Recursions (MoR) Bae et al. (2025), which introduces the concept of dynamic recursive depth for each token in large language models. MoR Bae et al. (2025) explores several routing strategies, including a "token-choice" mechanism where each token is statically assigned a fixed, absolute number of recursive steps to execute. However, in the context of PDE solving, the fluctuating number of active tokens across recursive levels can undermine both the stability and accuracy of the solution. In addition, this routing mechanism relies on auxiliary losses or routing biases during training, which may interfere with the model's ability to faithfully capture underlying physical laws.

## 3 MOTIVATION

In our research on solving PDE tasks, we have identified the following two phenomena.

### 3.1 NON-UNIFORM COMPLEXITY IN PDE TASKS

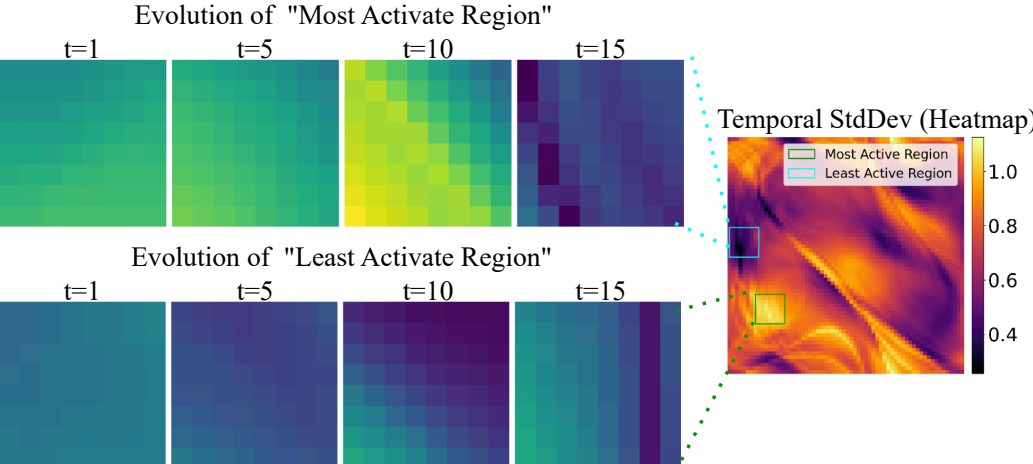

Figure 1: The heatmap on the right represents the intensity of change at this location across all time steps within the NS2D data; the two rows of images on the left respectively depict the temporal evolution of the most intense and most stable regions.

We observe that time-dependent systems governed by partial differential equations exhibit a fundamental property: their computational complexity is often highly localized in both space and time. Intensive computation is typically required only within sparse subregions undergoing dynamic evolution, while most of the domain changes only gradually. As shown in Figure 1, the image on the right depicts the degree of temporal variation in the NS2D dataset, highlighting that not all regions experience significant change.

## 3.2 CONCENTRATED ACTIVATIONS IN MODEL REPRESENTATIONS

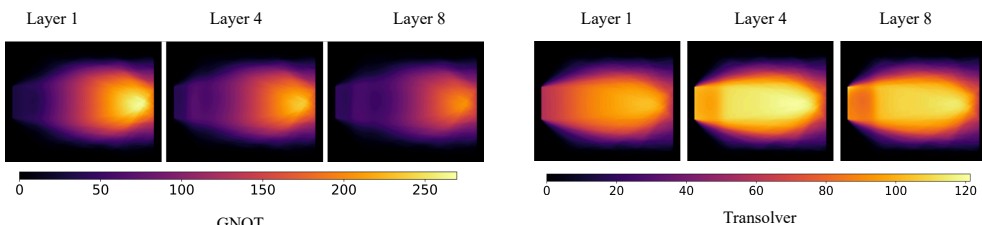

Figure 2: Transolver and GNOT model feature activation norms on the Pipe dataset, quantified as the L2 norm across all feature channels.

We sought to understand the performance of current Transformer-based neural operators on datasets with unevenly distributed information. To investigate this, we conducted a visual analysis of the internal states of two representative models, GNOTHao et al. (2023) and Transolver Wu et al. (2024), for the pipeline flow task. By aggregating the magnitudes of feature channels across different network depths, we measured the activation strength at each spatial location.

Figure 2 reveals a significant and consistent phenomenon present in both models: activation energy is strongly concentrated in physically critical regions starting from the outermost layer and persists throughout the entire network depth. We observe that even in the shallowest layers, activation intensity is highly concentrated in specific portions of the data. Furthermore, as depth increases, activation intensity becomes even more concentrated. This demonstrates that when training on such unevenly distributed data, not all regions of the model require processing through as many network layers. Therefore, we are motivated to design a new adaptive framework that learns a static importance map to guide the computational depth for each region.

## 4 METHODS

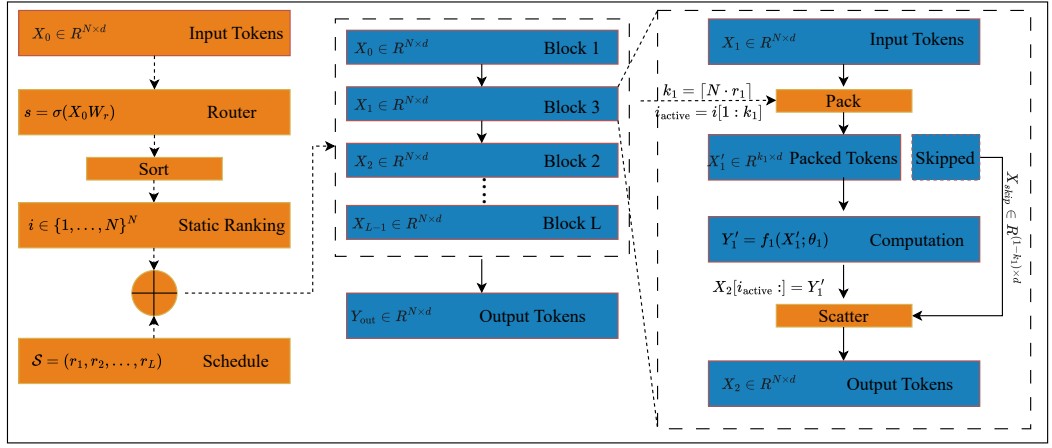

Figure 3: Flowchart of the Proposed SBR Method.

### 4.1 PROBLEM FORMULATION

Our primary goal is to learn the solution operator for a family of Partial Differential Equations (PDEs). Let $\mathcal{A}$ be a space of input functions (e.g., initial conditions, boundary conditions, or PDE coefficients) and $\mathcal{U}$ be a space of solution functions. We aim to learn a solution operator $\mathcal{G}^{\dagger} : \mathcal{A} \rightarrow \mathcal{U}$ that maps an input function $a \in \mathcal{A}$ to its corresponding solution $u \in \mathcal{U}$.

In practice, we work with discretized representations of these functions. An input function $a$ is represented by its values at $N$ points in the physical domain, $\{a(\boldsymbol{x}_i)\}_{i=1}^{N}$. These $N$ points are treated as a set of tokens, which are then projected into a $d$-dimensional feature space to form an input matrix $\boldsymbol{X}_{\text{in}} \in \mathbb{R}^{N \times d}$. A standard, $L$-layer Transformer-based neural operator, denoted as $F(\cdot; \boldsymbol{\theta})$, can be formulated as a function that maps this input matrix to an output matrix $\boldsymbol{Y}_{\text{out}} \in \mathbb{R}^{N \times d}$ representing the solution field:

$$\boldsymbol{Y}_{\text{out}} = F(\boldsymbol{X}_{\text{in}}; \boldsymbol{\theta}) = (f_L \circ f_{L-1} \circ \cdots \circ f_1)(\boldsymbol{X}_{\text{in}}). \tag{1}$$

where $f_l$ represents the $l$-th Transformer block parametrized by a subset of $\boldsymbol{\theta}$, and $\circ$ denotes function composition. In this standard formulation, each block $f_l$ processes the full set of $N$ tokens, leading to the uniform computation bottleneck.

The central challenge we address is to reformulate this process. Instead of applying each $f_l$ to all $N$ tokens, our adaptive framework learns a policy to select a subset of $k_l$ active tokens ($k_l \leq N$) for each layer $l$. The computation at layer $l$ is then performed only on this active subset. The goal is to significantly reduce the total computational cost, which scales with $\sum_{l=1}^{L} k_l$, while preserving the predictive accuracy of the operator.

## 4.2 THE GLOBAL ROUTER MODULE

The Global Router Module is the core component of SBR, responsible for realizing the "assessment" stage of our adaptive principle. It is a feed-forward network designed to perform a single, global analysis of the entire input field before the main processing begins. Its function is to quantify the relative importance of each discretized point, or token, in the domain, thereby generating a static computational plan. Specifically, you can refer to the leftmost section of the flowchart in Figure 3.

Formally, given the initial token embedding matrix $\boldsymbol{X}_0 \in \mathbb{R}^{N \times d}$, where $N$ is the number of tokens and $d$ is the feature dimension, the router module $R$ processes this input to generate a vector of scalar importance scores. This importance score vector $\boldsymbol{s} \in \mathbb{R}^N$ is computed as follows:

$$\boldsymbol{s} = \sigma(\boldsymbol{X}_0 \boldsymbol{W}_r). \tag{2}$$

Where $\boldsymbol{W}_r \in \mathbb{R}^{d \times 1}$ represents the learnable weights of the router's single linear layer, and $\sigma$ is the Sigmoid activation function. This operation effectively maps the high-dimensional representation of each token to a normalized, scalar value in $(0, 1)$, which serves as a proxy for its physical or computational importance.

These raw scores are then used to determine the static computational plan for the entire network. Specifically, we obtain a ranking vector $\boldsymbol{i} \in \mathcal{S}_N$, where $\mathcal{S}_N$ is the set of all permutations of $(1, \ldots, N)$, by sorting the tokens based on their importance scores $\boldsymbol{s}$ in descending order. This vector $\boldsymbol{i}$ contains the permuted indices of the original tokens, from most important to least important (i.e., $j$ is the index of the token with the $j$-th highest score). Crucially, this ranking $\boldsymbol{i}$ is generated only once and remains fixed throughout the entire forward pass of the backbone network. It serves as the definitive "computational roadmap" that dictates the processing priority for all subsequent layers.

## 4.3 THE ADAPTIVE PROCESSING BACKBONE

The Adaptive Processing Backbone is responsible for executing the static computational plan generated by the router. It utilizes the global importance ranking $\boldsymbol{i}$ to guide a structured, progressively sparse computational flow through the $L$ layers of the Transformer-based operator. The degree of sparsity at each layer is controlled by a pre-defined hyperparameter, the **Sparsity Schedule**.

**Sparsity Schedule.** The sparsity schedule is a user-defined sequence $\mathcal{S} = (r_1, r_2, \ldots, r_L)$, where $r_l \in (0, 1]$ is the ratio of tokens to keep active at layer $l$. This allows for flexible control over the computational budget. A typical schedule might be decremental, for instance, keeping all tokens active in the initial layers to capture global context ($r_l = 1.0$) and progressively reducing the ratio in deeper layers ($r_l < 1.0$) to focus computation.

**Structured Computational Flow.** For each layer $l \in \{1, \ldots, L\}$, the backbone performs an adaptive computation on the input token matrix $\boldsymbol{X}_{l-1} \in \mathbb{R}^{N \times d}$. This process, illustrated in the right of

Figure 3, begins by determining the number of active tokens for the current layer, $k_l = \lceil N \cdot r_l \rceil$. The indices of these active tokens, $\boldsymbol{i}_{\text{active}} \in \{1, \dots, N\}$, are then selected from the top of the global ranking vector, as $\boldsymbol{i}_{\text{active}} = \boldsymbol{i}[1 : k_l]$.

Next, a `pack` operation is performed to collect the features of these active tokens from $\boldsymbol{X}_{l-1}$ into a new, smaller, and dense matrix $\boldsymbol{X}'_l \in \mathbb{R}^{k_l \times d}$. All expensive computations of the Transformer block, such as self-attention and MLP, are then exclusively performed on this compact matrix:

$$\boldsymbol{Y}'_l = f_l(\boldsymbol{X}'_l; \boldsymbol{\theta}_l). \tag{3}$$

This is the key step where significant computational savings are realized. Finally, a `scatter` operation writes the updated features $\boldsymbol{Y}'_l$ back to a full-sized output matrix. To ensure that information from inactive (skipped) tokens is preserved, this is implemented via a residual connection: the output $\boldsymbol{X}_l$ is first initialized as a copy of the input, $\boldsymbol{X}_l = \boldsymbol{X}_{l-1}$, and is then updated only at the active indices:

$$\boldsymbol{X}_l[i, :] = \begin{cases} \boldsymbol{Y}'_l[j, :] & \text{if } i = \boldsymbol{i}_{\text{active}}[j] \text{ for some } j \in \{1, \dots, k_l\} \\ \boldsymbol{X}_{l-1}[i, :] & \text{otherwise.} \end{cases} \tag{4}$$

Crucially, since the `pack` and `scatter` operations are differentiable with respect to their inputs, the entire SBR framework can be trained end-to-end using a single loss function, without requiring any auxiliary balancing losses.

## 5 EXPERIMENTS

### 5.1 EXPERIMENTAL SETUP AND EVALUATION PROTOCOL

We conduct a comprehensive evaluation of our proposed SBR framework across a diverse range of benchmarks to validate its effectiveness and efficiency.

**Datasets and Baselines.** Our experiments are performed on these standard benchmarks **NS2D**, **Airfoil**,**Pipe** and **Heat2d**. We implement our SBR framework as a modular enhancement to a suite of state-of-the-art, multi-layer Transformer-based neural operators, including **OFormer**, **GNOT**, **Transolver**, and **IPOT** Li et al. (2022); Hao et al. (2023); Wu et al. (2024); Lee & Oh (2024). Our experiments with IPOT are conducted on the datasets supported by its official source code, which does not include configurations for the Pipe dataset. For brevity, we denote our enhanced models with an "SBR-" prefix (e.g., SBR-GNOT). Detailed descriptions of the datasets and baseline models are provided in Appendix B.1.

**Evaluation Metrics and Implementation Details.** We evaluate all models along two primary dimensions: accuracy, quantified by the relative L2 norm error(defined as equation 5), and efficiency, measured in FLOPs. To provide a fair and focused assessment of SBR's contribution, efficiency is reported with respect to the computational cost (FLOPs) of the operator backbone where SBR is applied. Complete implementation details, including all hyperparameters and the specific sparsity schedules used in each experiment, are provided in Appendix B.2.

$$\mathcal{L}_{L_2} = \frac{1}{|\mathcal{D}_{\text{test}}|} \sum_{\boldsymbol{u} \in \mathcal{D}_{\text{test}}} \frac{\|\hat{\boldsymbol{u}} - \boldsymbol{u}\|_2}{\|\boldsymbol{u}\|_2}. \tag{5}$$

### 5.2 MAIN RESULTS ON NEURAL OPERATOR

We now present the main results comparing our SBR-enhanced models against their original, dense counterparts across all four baseline operators and benchmark datasets. The comprehensive results, summarized in Table 1, demonstrate that SBR establishes a substantially improved accuracy-efficiency trade-off.

**Predictive Accuracy.** As shown in Table 1, SBR-enhanced models consistently achieve accuracy that is highly comparable to, and in several cases superior to, the original baselines. For instance, on the Pipe dataset, SBR-GNOT and SBR-OFormer both achieve a lower error than their dense counterparts, indicating that by focusing computation on critical regions, SBR can sometimes act

Table 1: Performance comparison of baseline models and our SBR method on three datasets. FLOPs (Floating-point Operations) is a metric that quantifies the total computational cost of a model or algorithm by counting the total number of basic arithmetic operations it requires to run. Relative L2 Error is reported in the upper row, FLOPs (GFLOPs) in the lower row.

| Model | Method | | Airfoil | Pipe | NS2D |
|---|---|---|---|---|---|
| Transolver | Baseline | Rel. L2 Error | 0.00455 | 0.00752 | 0.107 |
| | | FLOPs | 10.9 | 54.2 | 11.93 |
| | SBR | Rel. L2 Error | 0.00449 | 0.00747 | 0.173 |
| | | FLOPs | 7.94 | 27.5 | 7.05 |
| IPOT | Baseline | Rel. L2 Error | 0.0273 | – | 0.0453 |
| | | FLOPs | 6.22 | – | 0.0191 |
| | SBR | Rel. L2 Error | 0.0228 | – | 0.0352 |
| | | FLOPs | 3.186 | – | 0.0140 |
| Oformer | Baseline | Rel. L2 Error | 0.00884 | 0.00864 | 0.208 |
| | | FLOPs | 1.22 | 2.53 | 379 |
| | SBR | Rel. L2 Error | 0.00937 | 0.00751 | 0.229 |
| | | FLOPs | 0.845 | 1.75 | 138 |
| GNOT | Baseline | Rel. L2 Error | 0.00799 | 0.00550 | 0.177 |
| | | FLOPs | 6.59 | 38.6 | 76.9 |
| | SBR | Rel. L2 Error | 0.00823 | 0.00460 | 0.174 |
| | | FLOPs | 4.12 | 16.8 | 12.3 |

as a regularizer that improves generalization. On the Airfoil dataset, SBR-Transolver also shows a slight improvement. In other cases where a minor accuracy degradation is observed (e.g., SBR-GNOT on Airfoil), the difference is negligible, confirming that SBR's adaptive strategy successfully preserves the crucial information necessary for high-fidelity predictions. We do note an exception on the NS2D dataset, where SBR-Transolver shows a drop in accuracy. We attribute this to a unique interplay between the model's architecture and the dataset's nature. Specifically, the chaotic, long-range dependencies of NS2D turbulence may lead Transolver Wu et al. (2024) to adopt a highly diffuse, "non-compressible" representation. This makes the model inherently sensitive to token reduction on this specific task, highlighting that the optimal strategy for adaptive computation can be dependent on the interaction between a model's inductive biases and the problem's underlying physics.

**Computational Efficiency.** In exchange for this robust accuracy, SBR delivers dramatic and consistent reductions in computational cost, as detailed in Table 1. The reduction in backbone FLOPs is significant across all models and datasets. For example, on the computationally demanding Pipe and NS2D benchmarks, SBR reduces the FLOPs of GNOTHao et al. (2023) by 56 % and 84 %, respectively. Similarly, SBR-OFormer achieves a 63 % reduction on the NS2D task. Even for Transolver Wu et al. (2024) on Pipe, SBR cuts the computational load by nearly half (49%). As detailed in Appendix D.1, this substantial reduction in FLOPs also translates to practical end-to-end inference speedups, confirming the real-world benefits of our approach.

Overall, these results strongly validate the effectiveness of the SBR framework. By intelligently removing a large fraction of redundant computations from the network's backbone, SBR establishes a new, superior Pareto frontier for the accuracy-versus-efficiency trade-off in Transformer-based neural operators.

## 5.3 ANALYTICAL EXPERIMENTS

To provide deeper insights into the SBR framework's internal mechanics and design principles, we conduct a series of analytical and comparative experiments.

**Predictable Performance: SBR vs. MoR-style Routing.** We compare SBR's static routing with a MoR-style token selection baseline to highlight the advantage of our design in terms of computational predictability. In our MoR variant, the router deterministically assigns a fixed exit layer to each token. As shown in Figures 4 and 5, given the same total computational budget, the MoR strategy leads to substantial fluctuations in per-layer load. In sharp contrast, SBR enforces a predefined sparsity schedule, ensuring that the computational load is entirely deterministic and predictable. Furthermore, as illustrated in Table 2 and Table 3, SBR not only achieves higher accuracy than MoR Bae et al. (2025) but also delivers superior throughput.

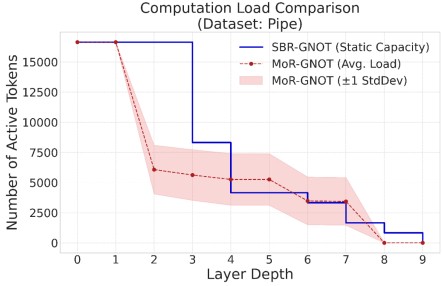

Figure 4: Load fluctuations of SBR and MoR under equal total capacity (Pipe)

Figure 5: Load fluctuations of SBR and MoR under equal total capacity (Airfoil)

Table 2: Mean Rel2 Error, Throughput, and Load Variance of SBR vs. MoR on Pipe

| Metric | SBR | MoR |
|---|---|---|
| Mean Rel2 Error(%) | 0.467 | 40.3 |
| Throughput (samples/s) | 70.73 | 53.63 |
| Avg Variance | 0.00e+00 | 2.54e+06 |

Table 3: Mean Rel2 Error, Throughput, and Load Variance of SBR vs. MoR on Airfoil

| Metric | SBR | MoR |
|---|---|---|
| Mean Rel2 Error(%) | 0.887 | 10.3 |
| Throughput (samples/s) | 102.97 | 97.17 |
| Avg Variance | 0.00e+00 | 2.04e+03 |

**Validation of the Core Hypothesis: Do Important Tokens Benefit More from Depth?** We conducted an experiment to validate SBR's core hypothesis: deeper layers provide greater benefits for physically complex regions. To this end, we partitioned the test set labels into "high-complexity" and "low-complexity" groups based on the ground-truth gradient magnitudes of the solution field—an objective, model-agnostic metric (see Appendix C). We then trained three standard dense GNOTHao et al. (2023) models from scratch—shallow, medium, and deep—and evaluated their prediction errors on both label groups. As shown in Figure 6, increasing the network depth from shallow to deep layers results in only modest improvements for low-complexity labels, especially when compared to the substantial gains observed for high-complexity labels. These findings provide strong empirical support for our hypothesis and validate SBR's strategy of allocating deeper computational resources selectively to regions where they yield the greatest returns.

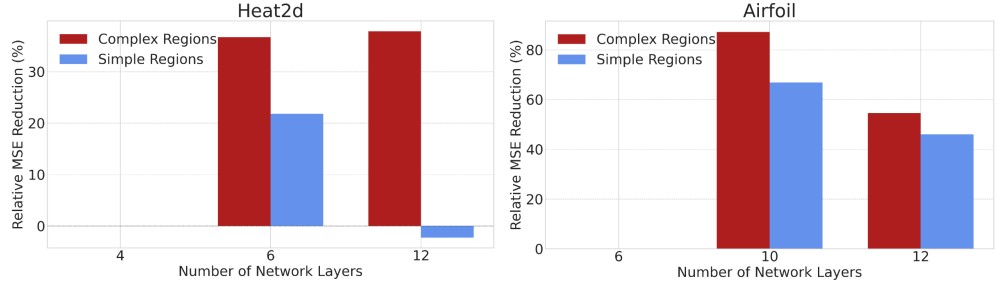

Figure 6: Relative loss reduction rate of standard GNOT in middle and deep layers compared with shallow layers on Heat2d and Pipe datasets.

Table 4: We compare our standard SBR-GNOT against a baseline with random routing under an identical computational budget on Pipe. Errors are reported as relative L2 norm (%).

| Model / Routing Strategy | Transolver (%) | GNOT (%) | OFormer (%) |
|---|---|---|---|
| SBR | **0.449** | **0.459** | **0.751** |
| Random | 0.539 | 0.483 | 0.784 |
| *Degradation* | *20.0%* | *5.2%* | *4.4%* |

## 5.4 ABLATION EXPERIMENTS

To validate the key design choices within the SBR framework, we conduct two targeted ablation studies. These experiments are designed to isolate and quantify the contribution of the learned routing mechanism and the structure of the sparsity schedule.

**The Importance of the Learned Routing Mechanism.** To isolate the contribution of our learned router, we conduct a crucial ablation study comparing the standard **SBR** with a **Random-SBR** baseline. This baseline employs the exact same decremental sparsity schedule, ensuring an identical computational budget (FLOPs), but replaces the importance ranking learned by the router with uniformly random token sampling. As shown in Table 4, replacing the learned router with random sampling results in a substantial drop in predictive accuracy across all backbone architectures. Specifically, the prediction error for Transolver Wu et al. (2024) increases dramatically by 20.0%, while OFormer Li et al. (2022) and GNOTHao et al. (2023) experience notable error increases of 4.4% and 5.2%, respectively.

This significant performance gap strongly demonstrates the critical importance of SBR's learned router to its effectiveness. Unlike the random SBR baseline, which distributes computation indiscriminately, even to physically quiescent regions, SBR's router identifies and prioritizes signatures in complex regions, such as shock fronts, vortices, and boundary layers. This targeted allocation of deep computation, consistent with the underlying physics, is the primary driver of SBR's efficiency and prediction accuracy. While all architectures benefit from intelligent routing, the extent of the performance degradation varies: Transolver Wu et al. (2024) experiences a performance drop of up to 20.0%, while OFormer Li et al. (2022) and GNOTHao et al. (2023) experience smaller but still significant drops, demonstrating the value of selective computation even for models with stronger local inductive biases, such as OFormer Li et al. (2022) and GNOTHao et al. (2023).

## 6 CONCLUSION

In this paper, we introduced Skip-Block Routing (SBR), a general and efficient framework designed to address the uniform computation bottleneck in Transformer-based neural operators. By decoupling complexity assessment from the main computational pipeline, SBR intelligently allocates deeper computational resources to physically critical regions while maintaining a hardware-friendly, static routing plan. Our extensive experiments demonstrate that SBR can be seamlessly integrated into various state-of-the-art operators, reducing computational cost by approximately 50% in terms of FLOPs and delivering end-to-end inference speedups of up to 2x, all without compromising predictive accuracy.

Looking forward, our work opens several exciting avenues for future research. While SBR's static routing ensures predictable performance, exploring routing mechanisms that are more deeply co-designed with the underlying physics of PDEs presents a compelling direction. For instance, developing hybrid strategies that combine a global plan with lightweight, dynamic adjustments could better handle problems with time-evolving phenomena like traveling shockwaves. Furthermore, designing routers that explicitly leverage known physical invariances or domain-specific knowledge could lead to even more robust and efficient allocation of computational resources. Ultimately, we believe that such physics-informed adaptive computation is a key step towards building not only accurate but also practical neural operators that can be deployed at scale for real-world scientific and engineering challenges.

## ETHICS STATEMENT

We have manually reevaluated the dataset we created to ensure it is free of any potential for discrimination, human rights violations, bias, exploitation, and any other ethical concerns.

## REPRODUCIBILITY STATEMENT

To ensure the reproducibility of our findings, all source code and datasets used in our experiments are included in the supplementary material. The provided materials are sufficient to replicate the main results presented in this paper.

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

## A    USAGE OF LLMs

Throughout the preparation of this manuscript, Large Language Models (LLMs) were utilized as a writing and editing tool. Specifically, we employed LLMs to improve the clarity and readability of the text, refine sentence structures, and correct grammatical errors. All final content, including the core scientific claims, experimental design, and conclusions, was conceived and written by us, and we take full responsibility for the final version of this paper.

## B    EXPERIMENTAL SETUP DETAILS

This appendix provides the detailed configurations used in our experiments to ensure full reproducibility.

### B.1    DATASET CONFIGURATIONS

All datasets used in our experiments are standard benchmarks adopted in prior neural operator literature, including FNO Li et al. (2020), Geo-FNO Li et al. (2023), and GNOT Hao et al. (2023), ensuring direct comparability with existing work. Each dataset consists of paired inputs and outputs that represent the parameters and corresponding solutions of a specific PDE system.

**NS2D (2D Navier–Stokes Turbulence).** This dataset simulates the evolution of a two-dimensional viscous, incompressible fluid in a periodic domain, governed by the Navier–Stokes equations. The task is to predict the vorticity field at a future time step from its initial condition. Each sample is discretized on a $64 \times 64$ grid. This benchmark typifies chaotic, time-dependent dynamical systems.

**Pipe.** This dataset models laminar fluid flow through a pipe with irregular geometry, governed by the steady-state Navier–Stokes equations. The task is to predict the velocity field inside the pipe from its boundary geometry. Each sample is discretized on a $129 \times 129$ grid. This benchmark emphasizes challenges in problems with complex but static geometries.

**Airfoil.** This dataset simulates steady-state, subsonic airflow around an airfoil, governed by the compressible Euler equations. The task is to predict the pressure and velocity fields given the airfoil shape. Each sample is discretized on a $221 \times 51$ grid. This benchmark reflects aerodynamic applications involving intricate boundary interactions.

**Heat2D.** This dataset involves solving the two-dimensional heat equation in a rectangular domain with spatially varying initial conditions. The task is to forecast the temperature distribution at a future time step from its initial profile. Each sample is discretized on a $64 \times 64$ grid. As a canonical diffusion benchmark, Heat2D highlights smooth temporal evolution and spatial propagation.

## B.2 BASELINE MODEL ARCHITECTURES

We apply our SBR framework to a suite of state-of-the-art Transformer-based neural operators. The core architectural parameters for each baseline model are detailed in Table 5.

Table 5: Model hyperparameter settings for different datasets.

| Model/Dataset | Pipe | NS2D | Airfoil |
|---|---|---|---|
| GNOT | n-hidden: 32
n-layer: 10 | n-hidden: 64
n-layer: 8 | n-hidden: 64
n-layer: 10 |
| Transolver | n-hidden: 128
n-layer: 8
n-head: 8 | n-hidden: 128
n-layer: 8
n-head: 8 | n-hidden: 64
n-layer: 8
n-head: 4 |
| OFormer | propagator_depth: 8 | propagator_depth: 10 | propagator_depth: 8 |
| IPOT | -
- | num_latents: 64
self_per_cross_attn: 6 | num_latents: 2048
self_per_cross_attn: 10 |

## C LOCAL COMPLEXITY QUANTIFICATION AND REGION PARTITIONING

### C.1 LOCAL COMPLEXITY QUANTIFICATION METRIC

For each discrete point $p_i$ in the solution field, we compute a scalar *complexity score* $s_i$ that measures the degree of local physical variability. Intuitively, the score reflects how rapidly the physical quantities change in the neighborhood of $p_i$.

**Gradient Magnitude.** The gradient measures the spatial rate of change of a field function. A larger gradient norm indicates more dramatic variation, thus suggesting a high-complexity region. Berger & Oliger (1984) For unstructured point cloud data, the gradient at $p_i$ is approximated as follows:

- **Neighborhood identification:** Use the $k$-nearest neighbors (k-NN) algorithm to locate the $k$ closest points to $p_i$ in coordinate space.

- **Local linear system:** Construct an overdetermined linear system $\Delta x \cdot \nabla f \approx \Delta y$, where $\Delta x$ are coordinate displacements relative to $p_i$ and $\Delta y$ are the corresponding differences in physical values.

- **Gradient estimation:** Solve the system using numerically stable least-squares to obtain the approximate gradient $\nabla f$ at $p_i$. Lancaster & Salkauskas (1981)

- **Score computation:** Define the complexity score as the L2 norm of the gradient vector:

$$s_i = \|\nabla f\|_2.$$

### C.2 QUANTILE-BASED REGION PARTITIONING

After computing the complexity scores $\{s_1, s_2, \ldots, s_N\}$ for all $N$ points in a sample, we partition the domain into regions via quantile-based thresholds. This ensures that the division adapts dynamically to different samples.

**Threshold determination.** We compute the lower quantile $q_{\text{low}}$ and the upper quantile $q_{\text{high}}$ (e.g., 75th percentile) of all scores in the sample.

**Region definitions.**

- **Simple Region:** $\{p_i \mid s_i < q_{\text{low}}\}$.
- **Complex Region:** $\{p_i \mid s_i > q_{\text{high}}\}$.
- **Moderate Region:** $\{p_i \mid q_{\text{low}} \leq s_i \leq q_{\text{high}}\}$.

# D SUPPLEMENTARY EXPERIMENTAL

This appendix provides supplementary results that support the conclusions presented in the main paper, including the end-to-end inference speedup analysis and the scalability analysis.

## D.1 END-TO-END INFERENCE SPEEDUP ANALYSIS

To demonstrate the practical benefits of SBR, we measure the end-to-end wall-clock inference time of our SBR-enhanced models against their dense counterparts.

The results in Table 6 demonstrate that, in most cases, SBR substantially reduces backbone FLOPs, leading to notable end-to-end speedups. For instance, GNOTHao et al. (2023) achieves a 4.46× acceleration on NS2D. However, Table 6 also reveals that SBR slows down the end-to-end performance of IPOT Lee & Oh (2024) on NS2D. This indicates that the effectiveness of SBR depends on the share of total computation time attributable to the model backbone. In models where the backbone dominates computational cost (e.g., deeper architectures or those with lightweight encoders/decoders), SBR yields more pronounced speedups.

Table 6: End-to-end wall-clock inference time comparison. Speedup is calculated as (SBR throughput / Baseline throughput).

| Dataset | Model | Speedup |
|---------|-------|---------|
| NS2D | OFormer | 1.84x |
| | GNOT | 4.46x |
| | IPOT | 0.94x |
| | Transolver | 1.36x |
| Pipe Flow | OFormer | 1.12x |
| | GNOT | 2.01x |
| | Transolver | 1.67x |
| Airfoil | OFormer | 1.02x |
| | GNOT | 1.46x |
| | Transolver | 1.24x |
| | IPOT | 1.41x |

## D.2 THE IMPACT OF THE SPARSITY SCHEDULE DESIGN.

We investigate the impact of computational budget allocation across network depth by designing an experiment with an 8-layer SBR-GNOT model under a fixed total budget. Four sparsity scheduling schemes are compared: Decremental, Constant, Incremental, and Mid-Heavy. While all schemes maintain the same total number of processed tokens across layers, they differ in how these tokens are distributed, allowing us to assess the effect of depth-wise allocation strategies on model performance:

- **Decremental :** A schedule that processes more tokens in early layers and fewer in deep layers. For example: $\{1.0\ 0.75\ 0.75\ 0.5\ 0.5\}$
- **Constant :** A uniform schedule that processes the same number of tokens at every layer. For example: $\{0.7\ 0.7\ 0.7\ 0.7\ 0.7\}$

- **Incremental :** An inverse schedule that processes fewer tokens in early layers and more in deep layers.For example: {0.5 0.5 0.75 0.75 1.0}

- **Mid-Heavy :** A schedule that concentrates computation in the middle layers.For example:{0.5 0.75 1.0 0.75 0.5}

Table 7: GNOT and Transolver performance under various capacity designs (same total capacity), measured by relative L2 Error.

| Model | Decremental | Constant | Incremental | Mid-Heavy |
|---|---|---|---|---|
| GNOT | 0.054 | 0.6433 | 0.0602 | **0.0518** |
| Transolver | 0.0056 | 0.3082 | 0.005 | **0.0045** |

The results, shown in Table 7, indicate that the Middle-Heavy schedule significantly outperforms the other strategies, highlighting the functional specialization of layers in Transformer-based operators for PDE modeling. Belinkov et al. (2018); Geva et al. (2020) The middle layers serve as a bottleneck for information exchange, where long-range dependencies and global couplings are primarily established. Tishby et al. (2000); Tishby & Zaslavsky (2015) Aggressive sparsity in this stage (as in the Constant and Incremental schedules) severely limits global interactions, leading to substantial performance degradation. In contrast, the shallow layers exhibit higher feature redundancy, likely focusing on local feature extraction, so a moderate reduction in capacity has minimal impact on model expressiveness. Belinkov et al. (2018); Geva et al. (2020) The deep layers primarily rely on residual propagation to refine and correct global representations, making them more robust to sparsification and allowing predictive stability even under reduced capacity. He et al. (2016) This analysis validates our choice of a Middle-Heavy schedule: by allocating more computational resources to the critical middle layers while leveraging redundancy in shallow and deep layers, our design achieves an optimal distribution of computational effort.

### D.3 SCALABILITY ANALYSIS OF ACCELERATION GAINS

To support our claim in Section 5.3 that SBR's practical benefits increase with model size, we investigate how the end-to-end speedup scales with network depth. We construct variants of GNOTHao et al. (2023) with different numbers of layers (6, 8, 10, and 12) and compare the speedup achieved by SBR-GNOT for each size.

As shown in Figure 7, the end-to-end speedup exhibits a clear upward trend as the number of layers increases. This is because in deeper models, the multi-layer backbone constitutes a larger fraction of the total computational cost, allowing the savings from SBR to have a more pronounced impact on the overall inference time. This result highlights the significant potential of SBR for future, large-scale neural operators where computational efficiency is paramount. At the same time, the Table 8 shows that SBR-GNOT consistently maintains competitive performance compared to GNOT across different network depths

Table 8: Relative L2 errors of GNOT and SBR-GNOT across different network depths.

| layer | GNOT | SBR-GNOT |
|---|---|---|
| 6 | 0.0599 | 0.0538 |
| 8 | 0.0558 | 0.0540 |
| 10 | 0.0536 | 0.0478 |
| 12 | 0.0537 | 0.0499 |

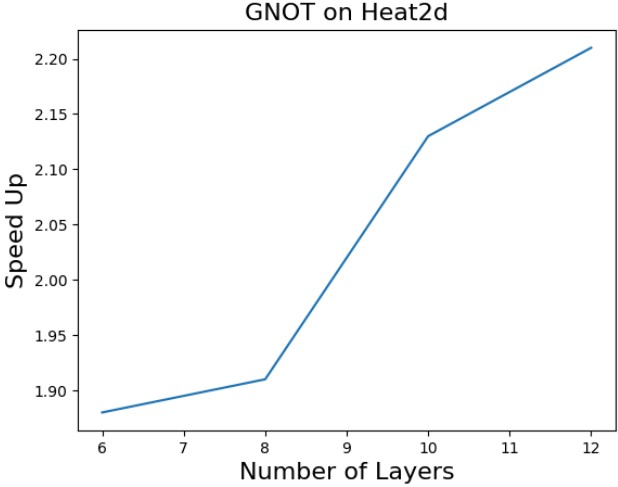

Figure 7: End-to-end speedup of SBR-GNOT versus the standard GNOT as the number of backbone layers increases. The speedup factor grows with model depth, demonstrating the scalability of SBR's benefits.

