# OpenReview forum: "From Uniform to Adaptive: General Skip-Block Mechanisms for Efficient PDE Neural Operators"
_ICLR.cc/2026/Conference — ICLR 2026 Conference Withdrawn Submission_

### Official Review · Reviewer_mnFo · 2025-10-27

**Soundness:** 3
**Presentation:** 1
**Contribution:** 3
**Rating:** 2
**Confidence:** 3

**Summary:**

This paper presents an adaptive computation framework for Transformer-based neural operators, which the authors term skip-block routing. A simple, single layer router ranks input tokens by "importance" or "complexity," assigning more computational resources to those tokens higher in the ranking. The top-ranked tokens are processed by deeper layers, as assigned by user-defined sparsity schedule. The end effect is a reduction in flops and (inference time as in the appendix) while maintaining the accuracy.

**Strengths:**

- The motivation is clear, as in numerical computing, we should focus computational resources on features which are difficult to model, e.g. sharp gradients.
- The method is simple and modular, easily being implemented within existing, widely used architectures.
- Efficiency gains in terms of FLOPS and speed are clear.
- Ablation studies, particularly with random routing and MoR, show that the token refinement strategy is actually helping to store useful information.

**Weaknesses:**

- I found issues with the presentation and clarity. There are quite a few typos and instances where the in-text citations are not formatted correctly. For example, \cite is used instead of \citep and there is no space between citations and other text. The exposition is also fairly dense and a bit repetitive, while concepts like physical intuition are buried in the details. Figures and tables aren't integrated into the narrative.
- The authors state the importance map is static and fixed. Is this fixed across samples? This seems poorly designed for time-dependent PDEs, especially with strongly nonlinear, local dynamics. This also seems counter to the whole point of the paper, to adaptively conform to the local complexities of PDEs.
- With the previous point, the time-dependent Navier-Stokes dataset shows very strong performance degradation. This is in contradiction to the author's statement that FLOPs may be decreased with no loss in accuracy.
- The paper lacks a systematic analysis of this approach for different types of PDEs. Some PDEs are globally dominated, while others are locally dominated. Without more investigation to a broader set of elliptic, parabolic, and hyperbolic PDEs, it is difficult to extrapolate the performance of the proposed method. Strong performance on a hyperbolic PDE would be very strong support for the authors' claims, but this is not investigated.
- I always recommend authors to provide a clear statement on the limitations of the current approach.

**Questions:**

- Could you clarify which types of PDEs your method is expected to perform well on? And do you expect it to perform well on hyperbolic or chaotic systems?
- As mentioned in the paper, the complexity of time-dependent systems is often localized in space and time. Some good examples which demonstrate this come from "the well" dataset's rayleigh_taylor_instability [https://arxiv.org/pdf/2412.00568 Figure 4], or the Kelvin Helmholtz and Curved Riemann problems in PDEGym [https://huggingface.co/collections/camlab-ethz/pdegym]. Can the authors provide a study applying one model to these problems? Perhaps transolver, as it gets the strongest results.
- How could SBR's fixed routing plan generalize to time-evolving or non-stationary problems, where local complexity is constantly changing?
- The importance score comes from a very simple network. Have you also tried more expressive models to calculate this score?
- Could you provide a systematic breakdown of the efficiency and accuracy tradeoffs for different physical regimes?
- Neural operators are notably resolution invariant. How does SBR perform under different resolutions?
- Is it possible to relate SBR's token pruning more explicitly to adaptive mesh refinement? The connection here seems natural, but it is not discussed in much detail.

---

### Official Review · Reviewer_FiLn · 2025-11-01

**Soundness:** 3
**Presentation:** 2
**Contribution:** 2
**Rating:** 4
**Confidence:** 4

**Summary:**

This paper proposes Skip-Block Routing (SBR), a model-agnostic framework that improves the efficiency of Transformer-based neural operators for solving PDEs. SBR introduces a routing mechanism that ranks token importance and adaptively allocates computation only to complex regions, reducing unnecessary processing. Benchmarks on several PDEs demonstrate that SBR maintains accuracy while reducing FLOPs by about 50% and achieving up to 2× faster inference.

**Strengths:**

1. The authors present a strong and well-motivated problem statement, clearly identifying the inefficiency arising from uniform computation in neural operators.
2. SBR is an appropriate and well-designed approach that effectively addresses the problem by introducing adaptive token-level computation.
3. The experimental evaluation is extensive, covering a wide range of architectures and PDE benchmarks.

**Weaknesses:**

1. The proposed method does not always outperform or comparable to the baselines. In Table 1, its performance is lower than existing models in many cases, which raises questions about the robustness and consistency of the approach.
2. The analysis in Figure 6 seems limited, since the comparison relies on a narrow experimental setting.

**Questions:**

Some suggestions:
1. In Figure 1, the color legend seems mismatched with the left-side evolution plots.
2. In Figure 3, the text inside the blue boxes is hard to read.
3. In the table 1,2 and 3, it would be easier to compare results if the best values were highlighted in bold.
4. Since the inference speedup is an important result mentioned in both the abstract and conclusion, it might be better to include the corresponding numbers from Appendix D (Table 6) directly in Table 1.

---

### Official Review · Reviewer_69bd · 2025-11-01

**Soundness:** 2
**Presentation:** 1
**Contribution:** 2
**Rating:** 2
**Confidence:** 4

**Summary:**

This paper proposes a general framework named "Skip-Block Routing (SBR)" designed to improve the computational efficiency of Transformer-based PDE neural operators. The SBR framework learns a static token-complexity ranking via a global router module; during inference, an adaptive backbone uses this ranking and a predefined "sparsity schedule" to process only a subset of tokens deemed more complex in deeper layers. The authors integrated SBR into several Transformer baselines, including OFormer, GNOT, Transolver, and IPOT, and evaluated it on benchmark datasets such as NS2D, Airfoil, Pipe, and Heat2d. The conclusions indicate that SBR can significantly reduce the model's computational cost and improve inference speed while maintaining predictive accuracy comparable to the original baselines.

**Strengths:**

1. The paper identifies the uniform computation bottleneck in existing Neural Operators, which is an important problem.
2. The authors' motivation to link the non-uniform complexity of physical fields to the model's computational resource allocation is reasonable.
3. The SBR framework is designed as a general module, and the paper demonstrates its integration across multiple Transformer-based operators.
4. The paper demonstrates, through experiments, the potential of SBR in reducing FLOPs.

**Weaknesses:**

1. The "static routing" design, which determines a fixed importance ranking based only on the initial state, is, in principle, incapable of capturing newly emerging complex regions in dynamic evolution problems. Concurrently, the "hard skip" mechanism copies skipped token features, thereby obstructing the global propagation of physical information (e.g., pressure, heat) through these "simple" regions and compromising physical fidelity.

2. The paper completely omits FLOPs and accuracy comparisons against established, efficient non-Transformer baselines in the field. This absence makes it impossible to judge the SBR framework's absolute value; the results can only demonstrate a relative improvement over existing Transformer models.

3. There is a significant disconnect between the paper's "large-scale engineering task" motivation and its experiments. All validations are confined to 2D, low-resolution datasets (e.g., $64 \times 64$ NS2D). No experiments are provided for 3D problems or high-resolution scenarios to substantiate these large-scale claims.

4. A core argument is that the SBR router "identifies critical regions". However, aside from analyzing the activations of baseline models in Figure 2, the paper provides no visualizations of what the SBR model itself learns to select versus skip. This leaves the core mechanism unverified, making it impossible to qualitatively assess whether it is learning physics or merely overfitting statistical artifacts.

5. Beyond the limited 4-strategy comparison in Appendix D.2, the paper severely lacks a sensitivity analysis on the token ratios ($r_l$) within the schedule $S$. For instance, there is no ablation on the absolute magnitude of these ratios, making it difficult to assess model robustness and the trade-off curve.

6. The overall presentation requires improvement.  Specifically, some citations in the paper are not formatted correctly. Additionally, the paper lacks qualitative case studies on any dataset.

**Questions:**

See Weaknesses.

---

### Note · Authors · 2025-12-08

I have read and agree with the venue's withdrawal policy on behalf of myself and my co-authors.